# Nanozybiotics: Nanozyme-Based Antibacterials against Bacterial Resistance

**DOI:** 10.3390/antibiotics11030390

**Published:** 2022-03-15

**Authors:** Caiyu Zhou, Qian Wang, Jing Jiang, Lizeng Gao

**Affiliations:** 1CAS Engineering Laboratory for Nanozyme, Institute of Biophysics, Chinese Academy of Sciences, Beijing 100101, China; zhoucaiyu17@mails.ucas.ac.cn (C.Z.); wangqian@ibp.ac.cn (Q.W.); jiangjing@ibp.ac.cn (J.J.); 2College of Life Sciences, Graduate School of University of Chinese Academy of Sciences, Beijing 100049, China; 3Nanozyme Medical Center, School of Basic Medical Sciences, Zhengzhou University, Zhengzhou 450001, China

**Keywords:** bacterial resistance, enzybiotics, nanozymes, nanozybiotics, biofilm

## Abstract

Infectious diseases caused by bacteria represent a global threat to human health. However, due to the abuse of antibiotics, drug-resistant bacteria have evolved rapidly and led to the failure of antibiotics treatment. Alternative antimicrobial strategies different to traditional antibiotics are urgently needed. Enzyme-based antibacterials (Enzybiotics) have gradually attracted interest owing to their advantages including high specificity, rapid mode-of-action, no resistance development, etc. However, due to their low stability, potential immunogenicity, and high cost of natural enzymes, enzybiotics have limitations in practical antibacterial therapy. In recent years, many nanomaterials with enzyme-like activities (Nanozymes) have been discovered as a new generation of artificial enzymes and perform catalytic antibacterial effects against bacterial resistance. To highlight the progress in this field of nanozyme-based antibacterials (Nanozybiotics), this review discussed the antibacterial mechanism of action of nanozybiotics with a comparison with enzybiotics. We propose that nanozybiotics may bear promising applications in antibacterial therapy, due to their high stability, rapid bacterial killing, biofilm elimination, and low cost.

## 1. Introduction

The diseases caused by bacterial infection represent a serious threat to human health. Antibiotics, as common chemical reagents against bacteria, have been widely used in human patients. However, the resistance to antibiotics is a growing and serious problem, with bacterial infection returning as a potential global threat. According to the Centers for Disease Control and Prevention (CDC), drug-resistant bacteria causes nearly 23,000 deaths per year on average in the United States [1]. The emergence of bacterial resistance is the result of its own multiple drug-resistant mechanisms, which greatly increases the difficulty of providing effective antibacterials and reduces the therapeutic effect of antibiotics. Therefore, it is necessary to develop new antimicrobial strategies while avoiding bacterial resistance.

In recent years, researchers have been devoted to discovering and developing new drug targets and antimicrobial methods. Common antimicrobial strategies include antimicrobial peptides, bacteriophage therapy, antimicrobial enzymes and so on. Antimicrobial peptides (AMPs) are generally polycationic amphiphilic peptides with less than 50 amino acids found in a wide variety of life forms [2]. The cationic AMPs display bactericidal action by disrupting the bacterial anionic cell membrane and by binding to intercellular targets, causing bacteria death [3]. Though AMPs are active against a broad spectrum of bacteria, the clinical translation of AMPs is limited by high manufacturing costs, unfavorable pharmacokinetics and cytotoxicity in vivo [4,5]. Bacteriophages (phages) are viruses that infect bacteria and propagate in their bacterial host, resulting in cell lysis. Some phages also produce depolymerases that hydrolyze biofilm extracellular polymers. Phage therapy shows great efficiency in lysing specific bacteria and degrading the biofilm matrix [6]. However, treatment of bacterial infections with phages bears some disadvantages, including narrow host range, bacterial resistance to phages and inactivation by the immune system [6,7]. Moreover, the phage cocktails strategy is limited by complicated procedures and strong individual specificity [8].

However, enzymatic therapy can overcome almost all obstacles mentioned above. Enzymatic therapy is a treatment that uses materials with catalytic activity to catalyze critical biochemical reactions, fulfilling therapeutic effects in vivo to treat various complications [9]. Enzymatic methods mainly include the application of natural enzyme-based antibacterials. Some natural enzymes, such as lysozyme or deoxyribonuclease, can hydrolyze the cell structure of bacteria and thus exhibit excellent antibacterial performance as enzymes-based antibacterials (Enzybiotics). They are highly specific, effective, and fast-acting. However, the proteinaceous character of enzybiotics results in low stability and potential immunogenicity, which limits their practical applications [10]. To overcome the limitations of proteinaceous enzymes, artificial enzyme mimics may represent a better choice. Recently, nanozymes, a kind of nanomaterials or nanostructures with enzyme-like activity, have attracted much attention as a new generation of enzyme mimics [11]. It has been reported that nanozymes with peroxidase-like or oxidase-like activity could produce a great number of reactive oxygen species (ROS) to fight bacteria [12,13,14,15]. Furthermore, nanozymes with DNase-like activity also prevent spread of drug resistance by degradation of bacterial drug resistance genes [16,17]. Typically, compared with natural enzymes, nanozymes show the characteristics of being more stable, economical, and practical [11]. Therefore, nanozyme-based antibacterial alternatives (Nanozybiotics) may have greater application potential in the antibacterial field. 

Herein, to understand and develop antibacterial strategies, we review the common mechanisms of bacterial drug resistance. Subsequently, the mechanisms and applications of enzybiotics and nanozybiotics as antibacterial agents, classified according to enzymatic activity, are summarized, including the combined strategies of nanozybiotics through enzymatic activities and specific physicochemical properties (e.g., photothermic or photodynamic therapy). We hope that this review can provide ideas for the development of a practical enzymatic therapy to combat the challenges of drug resistance in bacterial infections.

## 2. Non-Antibiotic Strategy Is Needed to Fight against Rapid Evolution of Bacterial Resistance

Antibiotics are considered one of the greatest achievements in the history of medicine, laying the foundation for the prevention and treatment of bacterial infection [18]. However, bacteria have also developed a variety of drug-resistant mechanisms to respond to them [19,20] (Figure 1). In order to maintain long-term effective antibacterial drugs, preventing the formation of super resistant bacteria, broadening the development avenues of antibacterial drugs, and a profound analysis of bacteria resistant mechanisms is necessary. Thus, here we briefly introduce the main mechanisms of bacterial drug resistance, including genetic, biochemical, and metabolic mechanisms.

Genetic mechanisms enact drug resistance through gene mutations leading to a change of drug target in bacteria which can survive treatment with antibiotics [19]. Moreover, other strain bacteria can acquire these drug-resistant genes through vertical transmission of chromosomes, horizontal transmission of plasmids or transposons or acquisition of exogenous drug-resistant genes by integrons [21]. Biochemical mechanisms are a more common style of antidrug. It mainly resists antibiotic treatment by target site mutation, antibiotic inactivation, impaired permeability of antibiotics, and efflux of antibiotics [22]. In addition, bacteria within a biofilm are a great challenge in the treatment of bacterial infection. Biofilms are bacterial self-generated protective communities, with an extracellular polymeric substance (EPS) matrix mainly composed of extracellular bacterial DNA (eDNA), exopolysaccharides, proteins and enzymes [23], and protects bacteria from attack by extraneous antibiotics. Besides, with the in-depth research on the mechanism of bacterial drug resistance, it has been noticed that the change in bacterial metabolic pathways could also affect susceptibility to antibiotics. To overcome drug resistance of bacteria, non-antibiotic strategies are needed to avoid the above issues. 

## 3. Enzybiotics Are Catalytic Antibacterials Based on Enzymes against Drug-Resistant Bacteria

The overuse and misuse of traditional antibiotics have led to the emergence of drug resistance worldwide. Hitherto, there is no current development of new antibiotic classes by pharmaceutical companies. Thus, it is urgent to develop non-antibiotic alternatives to fight against drug-resistant bacteria. Recently, enzybiotics represent a promising class of antibiotics alternatives. Enzybiotics are derived from natural enzymes with antibacterial performance, usually working by degrading the bacterial cell structure. They are characterized by a rapid and unique mode of action, a high specificity of killing pathogens, and a low probability for developing new bacterial resistance [10].

Enzybiotics with antibacterial effects are mainly classified into three categories: peptidoglycan hydrolases, proteases, and nuclease (antibacterial mechanisms shown in Figure 2). Firstly, peptidoglycan hydrolase is a general term for many kinds of enzymes which catalyze the hydrolysis of peptidoglycan, a major component of Gram-positive bacterial cell wall, and can be obtained from various sources, for example, bacteriophages (lysins or endolysins) and bacteria (bacteriocins and autolysins) [24]. The modular structure of lysins contains an N-terminal enzymatically active domain (EAD) connected via a flexible linker sequence to a C-terminal cell wall binding domain (CBD). The EAD is evolutionarily conserved and responsible for the peptidoglycan lytic activity of the enzyme [25]. As a bacteriocin whose antibacterial activity has been studied most thoroughly both in vitro and in vivo, lysostaphin eradicates planktonic and quiescent bacteria as well as inhibiting *Staphylococcus aureus* (*S. aureus*) growth in the biofilms as a result of hydrolyzing the penta-glycine cross bridges of *S. aureus* peptidoglycan. In another words, lysostaphin kills both dividing, non-dividing and encapsulated and unencapsulated *S. aureus* strains [26,27]. In addition, lysozymes are also an example of peptidoglycan hydrolase-based enzybiotics from hen egg and human. Though lysozymes exert poor killing effects on Gram-negative bacteria due to the thin layer of peptidoglycan protected by outer lipid membranes (Figure 2a) [28,29], additional modification can improve it [30,31]. Secondly, proteases are a kind of enzymes present in all forms of life, and degrade proteins by catalyzing the hydrolytic cleavage of the peptide chain. The major function of proteases in living organisms is the cleavage of proteins resulting in the degradation of damaged, misfolded, and potentially harmful proteins and therefore providing the cell with amino acids essential for the synthesis of new proteins. Recently, the pharmaceutical application of proteases was concerned. The purified protease from bacteria could be used for various purposes such as antibacterial activity against clinical pathogens as well as degrading slime and biofilms to limit bacteria. One of the most widely used proteases is subtilisin which is derived from *Bacillus* species. Subtilisin is a non-specific serine protease that provides the preferred cleavage on the carboxyl side of hydrophobic amino acid residues [32]. In addition, several plant-derived proteases possess antibacterial activity. For example, papain and bromelain showed bactericidal and effective inhibitory activity against a variety of pathogens [33,34]. Thirdly, nuclease, as the name suggests, are proteins that exhibit the enzymatic activity of degrading nucleic acids. They are very common and play an important role in biological activities. Moreover, the potential application of nuclease has also been considered in the antibacterial and antibiofilm fields. For example, Eller et al. devised a strategy that showed combination of RNase 1 and a secretory peptide, LL-37, displayed extraordinary antimicrobial activity [35]. Banu et al. investigated the anti-biofilm effect of marine bacterial DNase (MBD) by targeting the eDNA present in the biofilms of *Candida* spp. [36].

Enzybiotics show highly antibacterial properties owing to its rapid and unique mode of action and specificity of killing pathogens. Considering the working features of enzybiotics that destroy bacterial cell structures, enzybiotics not only kill metabolically active bacteria, but also those in a dormant state, even when embedded in biofilm (Figure 3). Moreover, this feature also ensures enzybiotics bear a low probability to develop bacterial resistance. These advantages indicate that enzybiotics are very ideal antibacterial agents. However, enzybiotics have a short half-life and raise neutralizing antibodies in vivo because of their proteinaceous nature, which seriously limits their clinical transformation. Moreover, the widespread application of enzybiotics in industrial processes is considered restricted due to poor stability of enzymes outside optimum operating conditions (pH, temperature, salts, surfactants) and high loading requirements [37]. Although immobilized enzymes and additional modification can improve the stability and antibacterial effect of enzybiotics to a certain extent [30,31,37,38], it increases the complexity and cost of preparation. Therefore, it might boost the catalytic therapy in antibacterial applications if the above problems are solved. 

## 4. Nanozybiotics Are Catalytic Antibacterials Based on Nanozymes with Enzyme-like Activities

Nanozymes are a new generation of artificial enzymes which have unique physicochemical properties and enzyme-like catalytic activity. They catalyze the substrate of natural enzymes under physiological conditions and follow similar enzymatic reaction kinetics [39,40,41,42]. The enzyme-like catalytic activity comes from the intrinsic nano-structure of the nanozyme itself, without the need to introduce additional catalytic functional groups or natural enzymes. 

Nanozyme was first reported in 2007 by Gao et al. [43] who found that the ferromagnetic–oxide nanoparticles (Fe_3_O_4_ MNPs) had an intrinsic peroxidase-like activity, could catalyze the peroxidase substrate 3,3,5,5-tetramethylbenzidine (TMB), *o*-phenylenediamine (OPD) and di-azo-aminobenzene (DAB), systematically studied from the perspective of enzymology characteristics of nanomaterials, established the response measurement standards, and used it as a substitute for natural peroxidase in immunoassay. Compared with natural enzymes and other artificial enzymes, nanozymes have better stability, lower cost and adjustable catalytic activity [44,45]. Taking inspiration from natural enzymes–enzybiotics as antibacterial reagents [46,47], we propose use of nanozybiotics to term nanozymes with antibacterial activity. Compared to enzybiotics, nanozybiotics have a broad spectrum and high durability, which is superior for tackling the challenge of antibiotic resistance of bacteria. Moreover, a variety of nanozybiotics have been developed for antibacterial application (partly summarized in Table 1). Therefore, in this section, the antibacterial mechanisms and applications of nanozybiotics based on nanozymes modeled by multiple enzyme-like activities are systematically introduced.

### 4.1. Peroxidase-like Nanozymes

Peroxidase (POD), such as horseradish peroxidase (HRP), is a kind of hemosiderin enzyme that catalyzes the decomposition of H_2_O_2_. It has many applications in wastewater treatment, environmental science, biosensing and organic synthesis. Since the peroxidase-like activity of Fe_3_O_4_ MNPs was first reported in 2007, other nanozymes with such activity have been discovered successively (Table 1), and the application of nanozymes has also been extended to many important fields including antibacterial treatment.

Peroxidases catalyze the decomposition of H_2_O_2_ and produce free hydroxyl radicals (·OH) with strong oxidation. Hydroxyl radicals are one of the most destructive reactive oxygen species (ROS). It not only decomposes nucleic acid, protein, polysaccharide, and other bacterial biofilm components, but also destroys the structural integrity of bacteria, leading to their death (Figure 4a,c) [122]. Typically, high concentrations of H_2_O_2_ (3% or more) are required for sterilization, but higher concentrations of H_2_O_2_ may cause cytotoxicity to normal physiological tissues [123]. In contrast, lower concentration of H_2_O_2_ (0.5% or less) is needed in the presence of the nanozyme to reach the great antibacterial efficacy [122,124]. So far, many nanozymes with peroxidase-like activity have been regarded as promising antibacterial nanozybiotics [59,110,125,126,127]. For example, iron–oxide nanozymes (IONzymes) were demonstrated to suppress the survival of intracellular *Salmonella Enteritidis* (*S. Enteritidis*) via its intrinsic peroxidase-like activity [125].

Moreover, Sun et al. showed that graphene quantum dots (GQDs) with peroxidase-like activity, in the presence of low concentration of H_2_O_2_, could produce a large amount of ·OH and effectively kill Gram-negative *E. coli* and Gram-positive *S. aureus*. Accordingly, GQDs-Band-Aids were designed to prevent wound infection and promote wound healing [128]. Li et al. prepared one efficient peroxidase-like Fe_3_C/N-doped graphitic carbon nanomaterial (Fe_3_C/N-C), which could enable the decomposition of H_2_O_2_ to ·OH, resulting in higher broad-spectrum antimicrobial activity than H_2_O_2_ alone. Compared with the HRP, Fe_3_C/N-C exhibited excellent catalytic activity in a wide range of pH (1.0–11.0) and temperature (25–70 °C) [56]. In this model, the Fe_3_C/N-C nanozyme endowed efficient antibacterial treatment (*E. coli* and *S. aureusas*) of wound infection in vivo even in the presence of a much lower-concentration of H_2_O_2_ (1.0 mM). Thus, accelerated wound healing in vivo can be achieved without using high concentrations of H_2_O_2_.

The antibacterial capacity of nanozybiotics is closely associated with their composition and nanostructure by adjusting enzyme mimic abilities [129]. Xi et al. [108] designed two types of copper/carbon nanozymes including two Cu states (Cu^0^ and Cu^2+^). They found that the copper/carbon nanozymes displayed multi-enzyme activities and their antibacterial mechanisms depended on Cu states. Hollow carbon spheres (HCSs) modified with CuO (CuO-HCSs) nanozymes would induce Gram-negative bacteria death (*E. coli* and *P. aeruginosa*) when releasing Cu^2+^. While Cu-HCSs nanozymes killing both Gram-positive (*Salmonella typhimurium*, *S. typhimurium*) and Gram-negative bacteria (*E. coli* and *P. aeruginosa*) were based on POD-like activity which is responsible for ROS generation.

Given the acidic microenvironment in bacterial accumulation due to large amounts of organic acid produced during anaerobic fermentation, Yu et al. [130] developed a novel acid-responsive ROS generator for biofilm removal by fabricating Zeolitic imidazolate framework-8 (ZIF-8) co-encapsulating the antibacterial ligand (lysine carbon dots, Lys-CDs) and targeted drug (folic acid, FA). The synthesized ZIF-8@Lys-CD@FA nanozybiotics showed peroxidase-like activity in an acid environment, and produced extremely active hydroxyl radicals to effectively destroy mature biofilms, resulting in the significantly improved bacteriostatic rate against *E. coli* and *S. aureus*. In addition, in the acidic environments, *S. aureus* and *E. coli* were more sensitive to ZIF-8@Lys-CD@FA with MIC values at 32 μg mL^−1^ and 62.5 μg mL^−1^ at pH 5.5, and 62.5 μg mL^−1^ and 125 μg mL^−1^ at pH 6.5, respectively. Moreover, the pH-dependent catalytic activity of nanozybiotics may limit their antimicrobial application under neutral pH. Therefore, the enhanced catalysis ability of citrate modified with Fe_3_O_4_ NPs has been developed by using ATP as a synergist [131]. In this work, Fe_3_O_4_ NPs exhibited superior antibacterial performance against *E. coli* and *Bacillus subtilis* (*B. subtilis*, Gram-positive) in the presence of H_2_O_2_ under a neutral pH environment with the assistance of ATP. 

Recently, catalytic nanozybiotics were shown to disrupt biofilms but lacked a stabilizing coating required for clinical applications. Dextran-coated iron–oxide nanozymes (Dex-NZM) [132] displayed strong peroxidase-like activity at acidic pH values, targeted biofilms with high specificity, and prevented severe caries without impacting surrounding oral tissues in vivo. The K_m_ values for H_2_O_2_ were found to be 27 μM and 2.5 mM for Dex-NZM and HRP, respectively. Besides, Dex-NZM/H_2_O_2_ treatment significantly reduced the onset and severity of caries lesions (vs control or either Dex-NZM or H_2_O_2_ alone) without adverse effects on gingival tissues or oral microbiota diversity.

### 4.2. Oxidase-like Nanozymes

Oxidase is an important enzyme that catalyzes redox reaction of oxygen. In oxidase-catalyzed reactions, the molecular oxygen (O_2_) is oxidized and converted to H_2_O or H_2_O_2_ (in certain cases to superoxide radicals, O_2_^−^) (Figure 4b) [40]. The active free radicals produced simultaneously in the catalytic process exert antibacterial activity. Due to its strong reactivity, H_2_O_2_ directly oxidizes the outer structure of bacteria, destroys the permeability barrier of bacteria, and leads to an imbalance of the electrochemical balance between internal and external substances of bacteria, resulting in bacterial death [47]. Similarly, O_2_^−^ can react directly with nucleic acids, proteins, etc., causing bacterial death. 

In particular, the oxidase- and peroxidase-like activities have shown to be major mechanisms of the antimicrobial efficiency for noble-metal-based nanozymes [133]. For example, Tao et al. [133] found firstly AuNPs as the oxidase mimic and then constructed the AuNPs supported on bifunctionalized mesoporous silica (MSN) (MSN-AuNPs) to achieve dual enzyme activities similar to those of peroxidase and oxidase. They evaluated the antibacterial potency and found that MSN-AuNPs not only exhibited striking antibacterial properties against both Gram-negative (*E. coli*) and Gram-positive (*S. aureus*) bacteria, but also degraded the existing biofilm and prevented formation of new biofilm efficiently. 

In addition, Ge Fang and co-workers found that palladium (Pd) nanocrystals exhibit facet-dependent oxidase- and peroxidase-like activities that endow them with excellent antibacterial properties [134]. In this work, Pd cubes with higher activities killed Gram-positive drug-resistant *Staphylococcus aureus* (*S. aureus*) and *Enterococcus faecalis*, while Pd octahedrons displaying stronger penetration into bacterial membranes exerted higher antibacterial activity for Gram-negative *Escherichia coli* (*E. coli*) and *Salmonella enteritidis*. Pd-based nanostructures also exhibit excellent oxidase-like activity [134]. It is reported that Pd@Ir octahedra demonstrates powerful bactericidal activity against both *E. coli* and *S. aureus* due to significant enhancement of oxidation byproducts [111]. While the application of noble-metals is limited due to high prices, a study reported that terbium oxide nanoparticles (Tb_4_O_7_ NPs) with oxidase-like activity at acidic pH values can be easily synthesized with low cost [110]. Moreover, Tb_4_O_7_ NPs were able to quickly oxidize a series of organic substrates and produce hydroxyl radicals. Not surprisingly, Tb_4_O_7_ NPs exhibited potent antimicrobial activity against both *S. aureus* and *E. coli*. When the concentration of Tb_4_O_7_ NPs increased to 100 μg/mL, nearly 90% of the *S. aureus* were killed. The application of the antibacterial activities of Tb_4_O_7_ NPs also were validated in a wound infection mouse model.

Recently, He et al. [135] synthesized bamboolike nitrogen-doped carbon nanotubes encapsulating cobalt nanoparticles (N-CNTs@Co) which showed 12.1 times higher oxidase-mimicking activity than that of the most reported CeO_2_. They demonstrated that N-CNTs@Co exerted a great antibacterial effect against Gram-positive (*S. aureus*) and Gram-negative (*E. coli*) in vitro and in vivo by catalyzing oxygen to produce a large number of ROS under acidic condition. Notably, during antibacterial experiments (20 days), neither *S. aureus* nor *E. coli* developed resistance to N-CNTs@Co.

All of the above indicate that nanozybiotics with oxidase-like activity bear great development potential in the field of antibacterial application. Moreover, given that oxidase can generate hydrogen peroxide, which is the substrate of peroxidase, it also suggested that the enzyme cascade reaction may be another choice for efficient antibacterial activity for nanozymes with multiple activities.

### 4.3. Deoxyribonuclease-like Nanozymes

Drug-resistant genes from dead bacteria can remain in the environment and spread to other microbes via horizontal gene transfer. Drug-resistant genes are essentially DNA, a complementary double strand composed of deoxynucleotide units, which are connected by 3′,5′-phosphodiester bonds. As the carrier of life genetic information, DNA is highly stable with a half-life of 521 years in the environment [136,137,138]. Even if ultraviolet, chlorine, ozone, and other physical methods are used, the destructive effect of DNA is not ideal [139,140,141]. However, DNA can be decomposed by natural nucleases, such as deoxyribonuclease (DNase), which acts on 3′,5′-phosphodiester bond (Figure 2c). Therefore, development of antimicrobial materials with the ability to degrade DNA can prevent the dissemination of released drug-resistant genes from dead bacteria (Figure 5a). In addition, considering that eDNA is an important part of biofilm, nanozybiotics based on nanozymes with DNase-like activity also demonstrate an antibiofilm effect (Figure 5b). Recently, artificial nucleases have been developed to mimic the hydrolytic cleavage of phosphodiester bond by taking advantage of multinuclear metal complexes, including transition metals and rare earth elements such as Cu(II), Cr(III), Zn(II), Ce(IV) [142,143,144,145,146]. Among them, cerium complex has attracted considerable attention due to its high catalytic efficiency and good biocompatibility [16,147].

Zhaowei Chen and co-workers designed a DNase-mimetic artificial enzyme (DMAE) for anti-biofilm application. DMAE effectively prevented more than 90% bacterial adhesion and inhibition by degrading eDNA in *S. aureus* extracellular polymeric substances (EPS) [16]. In addition, imidazolium type poly (ionic liquid) (PIL)/cerium (IV) ion-based electrospun nanofibrous membranes (PIL-Ce) showed a DNase-like activity [17]. The PIL-Ce was able to cleave the phosphodiester bond of BNPP and performed the characteristics of nucleases (K_m_ value = 0.2656 mM). The antibacterial test of PIL-Ce showed the high efficiencies to eradicate bacteria and disintegrate drug-resistant genes. The wound treatment test using MRSA infected mice as the model further demonstrated that PIL-Ce membranes combined both antibacterial and DNase-mimic properties, and may have potential application as a new “green” wound dressing to block the drug resistance spread in a clinical setting. Furthermore, rare earth elements and cerium can be complexed as artificial nucleases with a mechanism of action on the phosphodiester bond. Upon the addition of a Ce-containing complex, two adjacent Ce ions can interact with one phosphodiester bond, and render the phosphate bond susceptible to a nucleophilic species. 

### 4.4. Combination Therapy

Various antibacterial alternatives have been developed to overcome the drug-resistant behavior of microorganisms, such as peptides, metal–sulfide/oxides and carbon-based nanostructures [124,148,149]. However, they still suffer from biotoxicity, high costs, cumbersome preparation processes and/or pollution. Though nanozybiotics, possessing long-term storage, good stability, and tunable catalytic properties, could be applied to construct a range of antibacterial systems, there is still a large space to improve their antimicrobial performance [150]. Therefore, combination therapy seems to potentially be the better option to improve antibacterial effects as much as possible while minimizing side effects [151]. In various antimicrobial therapies, photo-activated strategies such as photothermal therapy (PTT), photodynamic therapy (PDT) have attracted widespread attention, owing to their less-invasive nature, low side effects, and good controllability compared to other antibacterial agents [152,153,154]. The antibacterial mechanism of combination therapy is summarized in Figure 6 with more details.

Recently, Yin et al. [155] reported that polyethylene glycol functionalized MoS_2_ nanoflowers (PEG-MoS_2_ NFs) with peroxidase-like catalytic activity and high photothermal conversion efficiency in the near-infrared (NIR) region combined the catalysis with PTT to provide a rapid and great antibacterial effect. In this work, PEG-MoS_2_ NFs could eliminate both Gram-negative ampicillin resistant *Escherichia coli* (Amp^r^ *E. coli*) and Gram-positive endospore-forming *Bacillus subtilis* (*B. subtilis*) with a low concentration of H_2_O_2_ and 808 nm irradiation. Such a combination not only improved the cell wall damage induced by ·OH, but also minimized the side effects of PTT with a large shortening the treatment time.

Zhang et al. [70] combined photodynamic, photothermal and peroxidase-like enzymatic activities to maximize the antibacterial efficiency by using oxygen-vacancy molybdenum trioxide nanodots (MoO_3−x_ NDs). The synergistic combination of multiple therapies enabled MoO_3−x_ NDs to possess a lower K_m_ value and a higher V_max_ value compared with other peroxidase mimics. In vitro antibacterial tests showed that MoO_3−x_ NDs/H_2_O_2_/808 nm NIR decreased viability of MRSA and ESBL-producing *E. coli*, by destroying the bacterial cell surface. In particular, with H_2_O_2_ at low concentration (100 μm) and upon NIR exposure, the MoO_3−x_ NDs was heated to the optimum enzymatic temperature of peroxidase-like activity and released utmost ·OH through the intrinsic photothermal effect for killing bacteria and accelerating wound healing.

Sun et al. [118] developed an antibacterial strategy by combining sonodynamic therapy and catalase-like activity of Pd@Pt nanozymes. Sonodynamic therapy (SDT) uses ultrasound to activate acoustic sensitizer and triggers ROS to produce antibacterial effects, which has the advantages of a non-invasive mode and good tissue penetration. A nanoplatform (Pd@Pt-T790) was constructed by bridging Pd@Pt nanoplates with the organic sonosensitizer meso-tetra(4-carboxyphenyl)porphine (T790). The modification of T790 onto Pd@Pt could offer Pd@Pt-T790 a “blocking and activating” enzyme-like activity, namely upon US irradiation, the nanozyme activity was effectively recovered to catalyze the decomposition of endogenous H_2_O_2_ into O_2_. This elaborate strategy was helpful to decrease the potential toxicity and side effects of nanozymes on normal tissues and had potential to realize active, controllable, and disease-loci-specific nanozyme catalytic behavior. Moreover, the anti-bacterial test of the Pd@Pt-T790-based SDT nano-system demonstrated therapeutic effects to eradicate methicillin-resistant *S. aureus* (MRSA)-induced myositis.

Collectively, compared with enzybiotics, nanozybiotics bear a greater and broader application prospect, which may overcome the limitations of natural enzymes and combine other antibacterial strategies easily (Table 2). In addition, with the characteristics of being more stable, economical, and practical, nanozybiotics are more suitable for widespread application in industrial processes. However, it is worth noting that the catalytic efficiency of nanozybiotics is still not very high compared with natural enzymes, and the types of enzyme activities are fewer. Moreover, it is interesting that enzybiotics usually kill bacteria by directly catalyzing the destruction of bacterial cell structure, whereas nanozybiotics realize antibacterial effects by producing toxic ROS to destroy the cell structure. Thus, to improve the antibacterial performance of nanozybiotics, developing a new catalytic activity and enhancing antibacterial targeting are necessary in future studies.

As an antibacterial strategy, the ultimate goal of nanozybiotics is in vivo application and clinical transformation. However, the current research mainly focuses on in vitro biological experiments or skin surface wound infection models [56,65,73,83,92,93,110], and there are only a few studies on antibacterial application of nanozybiotics in vivo [60,85,96,100,116]. For example, some nanozybiotics are designed to exert an excellent effect on *Helicobacter Pylori* (*H. pylori*) eradication in vivo via pH-responsive peroxidase-and oxidase-like activity [60,100,116]. Zhu et al. constructed a cationic chitosan coated ruthenium dioxide nanozyme (QCS-RuO2@RBT, SRT NSs) for the management of biofilm-associated infections, including chronic lung infection [85]. Although nanozybiotics bear biological enzyme activity and have shown excellent therapeutic efficacy in some animal models of in vivo diseases, most nanozymes are inorganic nanomaterials. Due to nanomaterials properties or function varies with the size, composition, morphology and shape, surface modification and surface charge, the studies of biological safety and toxicology is very complex, especially in clinical applications, more special attention and efforts need to be given. Moreover, it represents an important research direction in the field of nanozybiotics in the future.

## 5. Conclusions and Perspective

Enzymatic therapy represents a promising strategy to combat challenges posed by the drug resistance of bacteria. Natural enzyme-based enzybiotics, originate from nature with great environmental friendliness and exhibiting a high catalytic antibacterial ability. The antibacterial mechanism of enzybiotics usually describes the enzymatic degradation of the bacterial cell structure or biofilms leading to bacterial death. But their disadvantage is that most natural enzymes are unstable in the process of industrial production, which limits their large-scale application and increases costs. Although immobilization and additional modification can improve the stability of natural enzymes to some extent, it increases the production cost and the complexity of production operation. Nanozymes are a kind of nanomaterial with enzyme-like activity, and are economical and stable compared with natural enzymes. At present, nanozyme-based nanozybiotics have shown great antibacterial application prospects against resistant bacteria by mimicking enzyme-like activities of natural enzymes. Their antibacterial performance can be further improved by combining enzyme-like properties with other physiochemical properties of nanozymes such as PTT and PDT. In addition, most antibacterial tests were validated under in vitro environments or topically administered with in vivo models, which means there remains a long road before reaching clinical transformation. Thus, exploring new biocompatible nanozybiotics using enzyme-like nanozymes equipped with multiple antibacterial capabilities and applicable scenarios is of great importance. Altogether, we believe that nanozybiotics based on nanozymes with enzyme-like activity represent a new class of antibiotics alternatives.

## Figures and Tables

**Figure 1 antibiotics-11-00390-f001:**
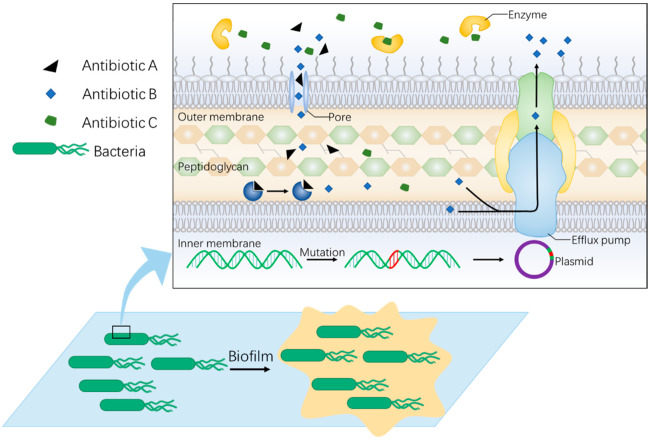
Mechanisms of bacterial resistance. The figure shows a brief overview of intrinsic resistance mechanisms. Firstly, bacteria acquire drug-resistance through gene mutation at the genetic level. The mutated gene can also spread through vertical and horizontal transmission (herein, plasmid for example). Besides, biochemical mechanisms are a more common style of antidrug upon target change, efflux or inactivation. Moreover, the formation of biofilm prevents bacteria from reaching the antibiotic and enhances drug resistance.

**Figure 2 antibiotics-11-00390-f002:**
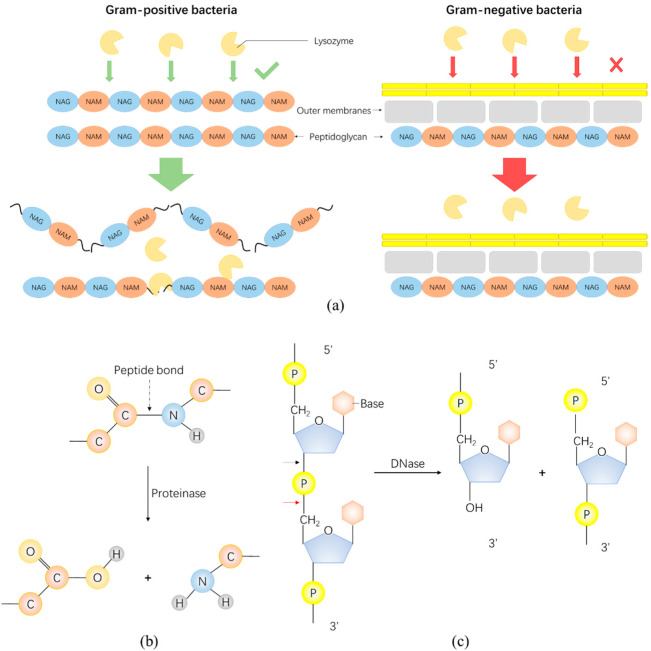
Schematic of the antibacterial mechanism of several typical enzybiotics. (**a**) Schematic of the action of the lysozyme on Gram-positive and Gram-negative bacteria. (**b**) Proteinase can degrade proteins by hydrolyzing peptide bonds. (**c**) Nuclease mainly consists of DNase and RNase, which degrade DNA and RNA, respectively. This figure uses DNase hydrolysis of phosphodiester bound as an example. The black arrow is the hydrolysis site of DNase and the red arrow is the hydrolysis site of RNase (e.g., RNase A family).

**Figure 3 antibiotics-11-00390-f003:**
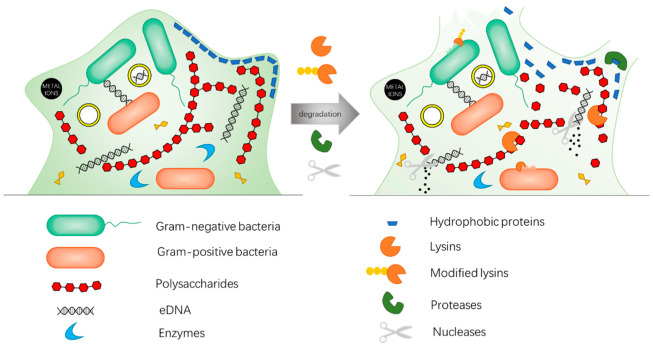
Mechanisms of enzybiotics eradicating bacterial biofilm. The biofilm consists of a combination of glycoproteins, carbohydrates, lipids, eDNA, and bacterial cell. When treating with enzybiotics, different enzymes bind to their respective targets for degradation. In this figure, lysins, proteases and nucleases bind to polysaccharides, proteins and eDNA in the biofilm and degrade them, respectively, thus promoting the disintegration of the biofilm. Besides, lysins also decompose the peptidoglycan, one of the important components of bacteria cell walls, to kill bacteria, especially for Gram-positive species. The peptidoglycan of Gram-negative bacteria is difficult to degrade due to the protection of the outer membrane, while modified lysins may take effect.

**Figure 4 antibiotics-11-00390-f004:**
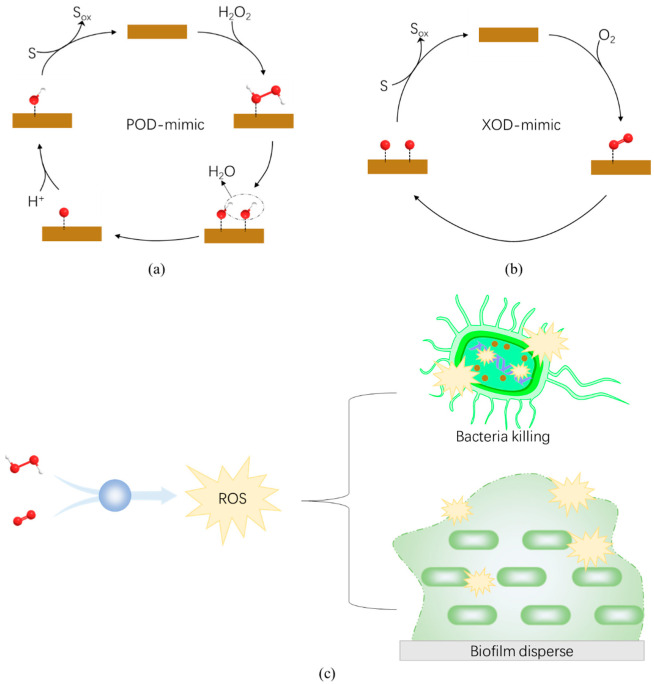
Schematic illustration of the catalytic reaction and antibacterial (and antibiofilm) mechanism of nanozybiotics based on nanozymes with peroxidase- or oxidase-like enzyme activities. (**a**) Nanozymes with peroxidase-like activity catalyze the reduction of H_2_O_2_ and produce free ·OH. (**b**) Nanozymes with oxidase-like activity catalyze O_2_ to ^1^O_2_, even single oxygen atoms. Both ·OH and ^1^O_2_ are strong oxidant, which oxidize the substrate (S) to ox-substrate (S_ox_), for example, membrane lipid. (**c**) Nanozymes with peroxidase- or oxidase-like activity destroy the membrane structure or degrade the biofilm matrix to kill bacteria.

**Figure 5 antibiotics-11-00390-f005:**
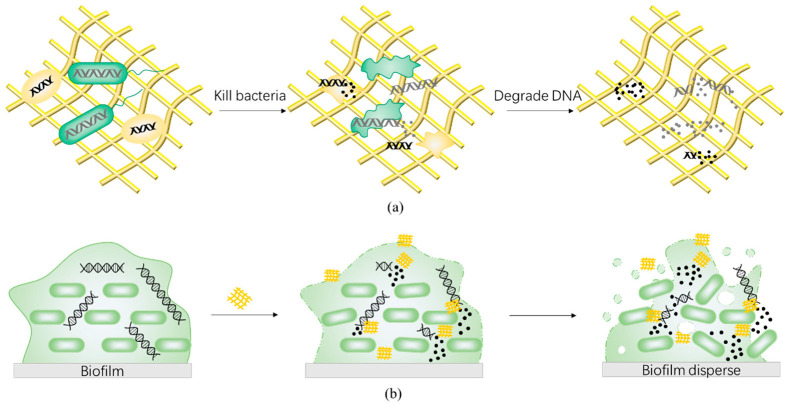
Schematic illustration of the antibacterial and antibiofilm mechanism of nanozybiotics based on nanozymes with deoxyribonuclease-like enzyme activities. Nanozymes with deoxyribonuclease activity catalyze the decomposition of DNA from dead bacteria to prevent the dissemination of released drug-resistant genes (**a**) and disperse the biofilm by decomposing the eDNA, the essential structural component of biofilm (**b**).

**Figure 6 antibiotics-11-00390-f006:**
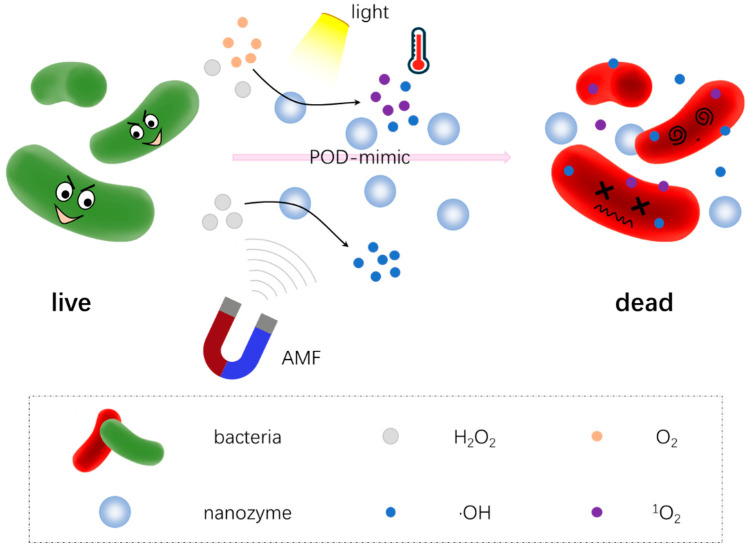
Schematic illustration of the antibacterial mechanism of combination therapy using nanozybiotics. Here, for example, nanozybiotics based on peroxidase (POD)-like nanozymes can catalyze the decomposition of hydrogen peroxide (H_2_O_2_) to produce bactericidal ·OH. With the radiation of light (visible or near-infrared light), photo-activated therapy (PTT and PDT) is active with excellent hyperthermia, singlet oxygen (^1^O_2_) and more ·OH production. The produced ·OH and ^1^O_2_ interact with bacteria to induce membrane peroxidation and damage cell integrity, making bacteria more vulnerable. In addition, the alternating magnetic field (AMF) exposure also can enhance the catalytic activity of nanozymes and generate more toxic ·OH.

**Table 1 antibiotics-11-00390-t001:** An overview of nanozybiotics as antibacterials.

Nanozybiotics	Enzyme Like Activity	Pathogens	Ref
PMCS	Peroxidase	*Pseudomonas aeruginosa* (*P. aeruginosa*)	[48]
2D Cu-TCPP(Fe) nanosheets	Peroxidase	*E. coli, S. aureus*	[49]
biohybrid CARs	Peroxidase	*Streptococcus mutans* UA159	[50]
MoS_2_-hydrogel	Peroxidase	*E. coli, S. aureus*	[51]
AA@Ru@HA-MoS_2_	Peroxidase	MDR *S. aureus, P. aeruginosa*	[52]
SAF NCs	Peroxidase	*E. coli, S. aureus*	[53]
hydrogel-based artificial enzyme	Peroxidase	Drug-resistant (DR) *S. aureus,* DR-*E. coli*	[54]
IONzymes	Peroxidase	*Streptococcus mutans (S. mutans)*	[55]
Fe_3_C/N-C	Peroxidase	*E. coli, S. aureus*	[56]
N-MoS_2_, N-WS_2_ NSs	Peroxidase	ampicillin resistant *Escherichia coli* (Amp^r^*E. coli),* endospore-forming *Bacillus subtilis (B. subtilis)*	[57]
N-SCSs	Peroxidase	MDR *S. aureus, E. coli*	[58]
UsAuNPs/MOFs	Peroxidase	*E. coli, S. aureus*	[59]
FNPs	Peroxidase	*Helicobacter pylori (H. pylori)*	[60]
CuMnO_2_ NFs	Peroxidase	*E. coli, S. aureus*	[61]
AIronNPs	Peroxidase	*E. coli, S. aureus*	[62]
NH_2_-MIL-88B(Fe)-Ag	Peroxidase	*E. coli, S. aureus*	[63]
IrNPs	Peroxidase	*E. coli*	[64]
CDs@PtNPs	Peroxidase	MRSA	[65]
Fe_3_O_4_@MoS_2_-Ag	Peroxidase	*E. coli, S. aureus, Bacillus subtilis (B. subtili), *MRSA, *Candida albicans (C. albicans)*	[66]
Au-BNNs, Ag-BNNs	Peroxidase	*E. coli, S. aureus*	[67]
oxygenated nanodiamonds (O-NDs)	Peroxidase	*Fusobacterium nucleatum (F. nucleatum), Porphyromonas gingivalis (P. gingivalis), S. sanguis*	[68]
Dex-IONP	Peroxidase	*S. mutans*	[69]
MoO_3−x_ NDs	Peroxidase	MRSA, ESBL-producing *E.coli*	[70]
MTex	Peroxidase	*E. coli, S. aureus*	[71]
Au/MnFe_2_O_4_	Peroxidase	*S. aureus, B. subtilis, E. faecalis, S. pyogenes*	[72]
AuNPTs	Peroxidase	MRSA, *E. coli, S. aureus*	[73]
rough C–Fe_3_O_4_	Peroxidase	MRSA, *E. coli, S. aureus*	[12]
Cu-PBG	Peroxidase	*E. coli, S. aureus*	[74]
PdFe/GDY	Peroxidase	*E. coli, S. aureus*	[75]
ultrasmall TA-Ag nanozyme	Peroxidase	*E. coli, Staphylococcus epidermidis (S. epidermidis)*	[76]
Cu-SA@BCNW/PNI	Peroxidase	*E. coli, S. aureus*	[77]
PEG@Zn/Pt–CN	Peroxidase	*E. coli, S. aureus*	[78]
Fe-N-C SAzyme	Peroxidase	*E. coli, S. aureus*	[79]
SA-Pt/g-C_3_N_4_-K	Peroxidase	*E. coli, S. aureus, Bacillus cereus, P. aeruginosa*	[80]
PDA/Fe_3_O_4_	Peroxidase	*E. coli, S. aureus*	[81]
CuFeSe_2_	Peroxidase	*S. aureus*	[82]
pFe_3_O_4_@GOx	Peroxidase	*E. coli, S. aureus*	[83]
Cu SASs/NPC	Peroxidase	*E. coli,* MRSA	[84]
QCS-RuO_2_@RBT	Peroxidase	*P. aeruginosa*	[85]
FerIONP	Peroxidase	*S. mutans*	[86]
*w*-SiO_2_/CuO	Peroxidase	*E. coli*	[87]
PdCu-Urchin	Peroxidase	*E. coli, S. aureus*	[88]
Au@Cu_2−x_S NPs	Peroxidase	*E. faecalis, Fusobacterium nucleus*	[89]
Cu_2_WS_4_ nanocrystals (CWS NCs)	Peroxidase, oxidase	MDR *S. aureus, E. coli*	[90]
3CoV-400	Peroxidase, oxidase	*E. coli, Bacillus algicola, Staphylococcus sciuri (S. sciuri), Vibrio harveyi, Pseudoalteromonas*	[91]
VO_x_NDs	Peroxidase, oxidase	*E. coli, S. aureus*	[92]
GO NSs, CuS/GO NC	Peroxidase, oxidase	*E. coli, S. aureus,* MRSA	[93]
Co_4_S_3_/Co_3_O_4_ NTs	Peroxidase, oxidase	*E. coli, S. sciuri*	[94]
Cu_2_MoS_4_ NPs	Peroxidase, oxidase	MDR *E. coli,* MDR *S. aureus*	[95]
WS_2_QDs	Peroxidase, oxidase	Mu50 (a vancomycin-intermediate *Staphylococcus aureus* reference strain)*, E.coli*	[96]
Pd@NPs	Peroxidase, oxidase	*S. aureus, P. aeruginosa*	[97]
NiCo_2_O_4_-Au	Peroxidase, oxidase	*E. coli, S. aureus*	[98]
CS-Cu-GA NCs	Peroxidase, oxidase	*E. coli, S. aureus*	[99]
MSPLNP-Au-CB	Peroxidase, oxidase	*Helicobacter pylori* (*H. pylori*)*,* MRSA	[100]
CSG-M*_X_*	Peroxidase, oxidase	*E. coli, S. aureus*	[13]
Cu_2−x_S	Peroxidase, oxidase	Amp^r^*E. coli* ^1^*, S. aureus*	[101]
CuO NPs/AA	Peroxidase, oxidase	*E. coli, S. aureus*	[102]
HvCuO@GOx	Peroxidase, catalase	*E. coli, S. aureus, E. coli* with streptomycin resistance (SR-E. *coli*)	[103]
FePN SAzyme	Peroxidase, catalase	*E. coli, S. aureus*	[104]
Au-Au/IrO_2_@Cu (PABA)	Peroxidase, glucose oxidase (GOx)	*E. coli, S. aureus*	[105]
Ti_3_C_2_ MXene/MoS_2_ (MM) 2D bio-heterojunctions	Peroxidase, glutathione oxidase	*E. coli, S. aureus*	[106]
MoS_2_/rGO VHS	Peroxidase, oxidase, catalase	*E. coli, S. aureus*	[107]
Cu-HCSs, CuO-HCSs	Peroxidase, catalase, superoxide dismutase	Cu-HCSs: Gram-positive and negative bacteria (*S. aureus, S. typhimurium, E. coli, P. aeruginosa*) CuO-HCSs: Gram-negative bacteria (*S. typhimurium, E. coli, P. aeruginosa*)	[108]
CNT@MoS_2_ NSs	Peroxidase, superoxide, catalase	*E. coli, S. aureus*	[109]
MoS_2_-PDA nanozyme composite hydrogel (MPH)	Peroxidase, catalase, superoxide dismutase	*E. coli, S. aureus*	[15]
Tb_4_O_7_ NPs	Oxidase	*S. aureus, E. coli*	[110]
Pd@Ir octahedra (or cubes)	Oxidase	*E. coli, S. aureus, Bacillus subtilis, Salmonella enteritidis*	[111]
Co_4_S_3_/Co(OH)_2_ HNTs	Oxidase	*E. coli, P. aeruginosa, S. sciuri, Bacillus*	[112]
Mn/Ni(OH)_x_ LDHs	Oxidase	*E. coli, S. aureus*	[113]
SPB NCPs	Oxidase	*S. aureus, P. aeruginosa*	[114]
AgPd_0.38_	Oxidase	*S. aureus, B. subtilis, E. coli, P. aeruginosa,* MRSA	[115]
PtCo@Graphene	Oxidase	*H. pylori*	[116]
MoS_2_/TiO_2_ NFs	Oxidase	*E. coli, S. aureus*	[117]
Cu_3_/ND@G	Oxidase	*E. coli*	[14]
Pd@Pt-T790	Catalase	MRSA	[118]
DMAE	DNase	*S. aureus*	[16]
PIL-Ce	DNase	*E. coli, S. aureus,* MRSA	[17]
CeO_2−x_ nanorods	Haloperoxidase	*E. coli*	[119]
Ce_1−x_Bi_x_O_2−δ_	Haloperoxidase	*P. aeruginosa, Phaeobacter gallaeciensis*	[120]
Cu-HCSs	Nuclease/protease	MRSA	[121]

^1^ Amp^r^ *E. coli*: Ampicillin-resistant *E. coli*.

**Table 2 antibiotics-11-00390-t002:** Comparison between enzybiotics and nanozybiotics for antibacterial application.

	Enzybiotics	Nanozybiotics
Derivation	natural enzymes	nanozymes (nanomaterials)
Catalytic activity	peptidoglycan hydrolases, proteases, and nuclease	peroxidase, oxidase, catalase, deoxyribonuclease
Main antibacterial mechanism	destroy bacterial cell structure	catalyze the production of ROS
Application advantages	rapid and unique mode of action, high specificity of kill pathogens, low probability for bacterial resistance development and a proteinaceous nature	economical, stable, with catalytic function without additional modification, easy to integrate a variety of antibacterial strategies
Application disadvantages	environmentally sensitive and unstable, high cost, short half-life and immunogenicity of proteins	low enzyme activity, limited types of enzyme catalysis and complicated toxicological profile

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
