# Peer review of "Nanozybiotics: Nanozyme-Based Antibacterials against Bacterial Resistance"

_antibiotics, 2022, doi:10.3390/antibiotics11030390_

Round 1

Reviewer 1 Report

The review article “Nanozybiotics: Nanozyme-based antibacterials against bacterial resistance” highlights a very important topic. The figures in the article are very informative and interesting, but I have following comments,

  1. “Non-Antibiotic Strategy Is Needed to Fight against Rapid Evolution of Bacterial Resistance” section line 84, the reference is missing and it is showing an error.
  2.  “Enzybiotics Are Catalytic Antibacterials Based on Enzymes against Drug-Resistant Bacteria” line 123 the reference is missing and it is showing an error.
  3. Lines 139, 170, 219, 226 and 291, the reference is missing and it is showing an error.
  4. lines 343-348 are showing reference errors, please check them.
  5. Reference source is missing in at line 388 & 433.
  6. It is not clear how the literature search for this review was carried by the authors? Which search strategy was used? How the article was included and what was the exclusion criteria?

Reviewer 2 Report

Dear Authors,

Your manuscript entitled “Nanozybiotics: Nanozyme-based antibacterials against bacterial resistance”  is really interesting and deals with a captivating topic, providing a clear and well-presented overview of the promising applications of nanozymes in antibacterial therapies. Therefore, I recommend the manuscript for publication in Antibiotics journal after minor revisions.

  • The authors presented the application of nanomaterials with peroxidase/oxidase activities, such as palladium and ferromagnetic oxide nanoparticles, as innovative antibacterial agents. Among nanozymes, platinum nanoparticles have recently emerged for their strong peroxidase-like activities. What’s about the application of this nanomaterial as antibacterial agent?
  • What is the toxicological profile of the nanomaterials presented? The authors should better address this aspect also considering the in vivo applications disadvantages of enzybiotics (eg. immunogenicity).
  • The text presents some typos and other errors (e.g. Error! Reference source not found).

All the best

Referee

Author Response

Reviewer 2

Dear Authors,

Your manuscript entitled “Nanozybiotics: Nanozyme-based antibacterials against bacterial resistance”  is really interesting and deals with a captivating topic, providing a clear and well-presented overview of the promising applications of nanozymes in antibacterial therapies. Therefore, I recommend the manuscript for publication in Antibiotics journal after minor revisions.

Response: We appreciate the positive comments of the reviewer.

  1. The authors presented the application of nanomaterials with peroxidase/oxidase activities, such as palladium and ferromagnetic oxide nanoparticles, as innovative antibacterial agents. Among nanozymes, platinum nanoparticles have recently emerged for their strong peroxidase-like activities. What’s about the application of this nanomaterial as antibacterial agent?

Response: The platinum nanoparticles with strong peroxidase activity showed effective antibacterial activity toward E. coli, S. aureus, MRSA, P. aeruginosa. The information has been summarized in Table 1, ref. [65], [78], [80].

  1. What is the toxicological profile of the nanomaterials presented? The authors should better address this aspect also considering the in vivoapplications disadvantages of enzybiotics (eg. immunogenicity).

Response: Thanks for pointing out this problem. Due to the properties or functions of nanomaterials change with size, composition, morphology and shape, surface modification and surface charge, the profile of biosafety or toxicity is very complicated. Some nanomaterials composed with noble metals may cause long term toxicity as they are not metabolizable in biological system, which is suitable for in vitro or topical (e.g. skin) applications. Those with biocompatible composition, such as iron oxide, may be more suitable for in vivo antibacterial treatment. However, since this field is just emerging recently, the major studies focused on in vitro or topical administration and the biosafety concern will be considered as a critical issue in the near future. We have also added this to “Application Disadvantages of Nanozybiotics” in Table 2 of the manuscript.

  1. The text presents some typos and other errors (e.g. Error! Reference source not found).

Response: We have carefully checked the problems and fixed them.

Reviewer 3 Report

In this submission, the author has discussed alternative antimicrobial therapy with Enzybiotics, Nanozymes, and Nanozybiotics to harness the antibacterial resistance. The review discusses the different mechanisms of action for nanozybiotics and enzybiotics. The author proposed that nanozybiotics-based antibacterial therapy would be one of the promising therapies, due to their high stability, rapid bacterial killing, biofilm elimination, and low cost of preparation.

The manuscript has been well written and could be published after considering the comments and suggested modifications.

Here are my comments which need to be addressed.

  1. Although accumulating evidence based on in vitro results says that synthetic nanozybiotics have a potent bactericidal effect, however, how these nanomaterials are selective towards the bacterial cells, in contrast with normal cells?
  2. It would be good to add one or two paragraphs focusing more on the outcome of the in vivo results of nanozybiotics which would be of great interest to the reader.
  3. Would there be any effect of surface functional groups on the antibacterial activity of synthetic nanozybiotics?
  4. It would be good to mention all stimuli in figure 6. For example, the temperate symbol is there but not designated by word.
  5. Page 1, line 40:

"cationic AMPs display bactericidal action by disrupting the bacterial anionic cell membrane"

To support this statement, the following literature could be incorporated in this review.

The literature Title: Coordination-Assisted Self-Assembled Polypeptide Nanogels to Selectively Combat Bacterial Infection.

  1. Page 2, line 79:

"Non-Antibiotic Strategy Is Needed to Fight against Rapid"

The ‘I’ of the word ‘Is’ should be a small letter as 'is' and a similar issue with the word ‘against’. These should be taken care of.

  1. Page 2, line 83: "the abuse and misuse of antibiotics, bacteria have also developed a variety of drug-resistant mechanisms "

This statement is repeated many times in the manuscript (Abstract, Introduction, and in section 2 as well). It would be good to present it in a slightly different way.

  1. Page 4, line 132: ‘In Vivo’ should be Italic.

  1. In many places of the review, it appears that “Error! Reference source not found”. It should be considered to fix before publication.

Author Response

Reviewer 3

In this submission, the author has discussed alternative antimicrobial therapy with Enzybiotics, Nanozymes, and Nanozybiotics to harness the antibacterial resistance. The review discusses the different mechanisms of action for nanozybiotics and enzybiotics. The author proposed that nanozybiotics-based antibacterial therapy would be one of the promising therapies, due to their high stability, rapid bacterial killing, biofilm elimination, and low cost of preparation.

The manuscript has been well written and could be published after considering the comments and suggested modifications.

Response: We appreciate the positive comments of the reviewer.

Here are my comments which need to be addressed.

  1. Although accumulating evidence based on in vitro results says that synthetic nanozybiotics have a potent bactericidal effect, however, how these nanomaterials are selective towards the bacterial cells, in contrast with normal cells?

Response: Thanks for the reviewer’s critical question. Currently most nanozybiotics are nanozymes with peroxidase-like or oxidase-like activities which arise ROS level to kill bacteria. However, it has been known that high ROS level may damage host cells. Our recent studies demonstrated that nanozymes cause specific lipid peroxidation to kill bacteria, resulting a ferroptosis-like death of bacteria[1]. Importantly, the sensitivity of bacteria to lipid ROS is 20-fold higher than that of host cells. Thus by this way, low dosage of nanozymes can kill bacteria and avoid to damage normal cells.

Furthermore, some nanozybiotics have environmental responsive property. For example, nanozymes with peroxidase-like activity start to work once reaching acidic microenvironment, and such activity can be turned on with light irradiation. Our recent studies found that some nanozymes (e.g. iron sulfide nanozymes) also can release polysufide species to selectively kill Gardnerella vaginalis by inhibiting bacterial glycolysis[2]. Such property may contribute to the selective antibacterial. In addition, nanozymes can be modified with certain ligands that bind to bacteria and specifically kill bacteria. For future study, we believe that nanozymes with lysozyme-like activity, protease-like activity or esterase-like activity will be developed. These nanozymes may be used to specifically digest bacterial wall rather than to damage normal cells. We believe that with the development and optimization of nanozybiotics, the antibacterial selectivity will be resolved gradually by combining the cutting-edged technologies in nanomaterials deign and synthesis.

  1. It would be good to add one or two paragraphs focusing more on the outcome of the in vivo results of nanozybiotics which would be of great interest to the reader.

Response: Thanks for such constructive suggestion. Following your suggestion, we have added a new paragraph to introduce the outcome of the in vivo results of nanozybiotics in the end of “4.4 combination therapy” subsection as well as below.

As an antibacterial strategy, the ultimate goal of nanozybiotcis is in vivo application and clinical transformation. However, the currently researches mainly focus on in vitro biological experiments or skin surface wound infection model[3-9], and there are only a few studies on antibacterial application of nanozybiotcs in vivo [10-14]. For example, some nanozybiotics are designed to exert an excellent effect on Helicobacter Pylori (H. pylori) eradication in vivo via pH-responsive peroxidase-and oxidase-like activity [11-13]. Zhu et al constructed a cationic chitosan coated ruthenium dioxide nanozyme (QCS-RuO2@RBT, SRT NSs) for the management of biofilm-associated infections, including chronic lung infection[14]. Although nanozybiotics have biological enzyme activity and have shown excellent therapeutic efficacy in some animal models of in vivo diseases, most nanozymes are inorganic nanomaterials. Due to nanomaterials properties or function varies with the size, composition, morphology and shape, surface modification and surface charge, the studies of biological safety and toxicology is very complex, especially in the clinical application, more need to be given special attention and efforts, it is also an important research direction in the field of nanozybiotics in the future.

  1. Would there be any effect of surface functional groups on the antibacterial activity of synthetic nanozybiotics?

Response: Thanks for the constructive question. We think that surface functional groups will affect the antibacterial activity of synthetic nanozybiotics by regulating their affinity to substrates. There are some studies to prove this. For example, Zhang et al fabricated fullerenol nanoparticles (FNPs) with varied chemical structures responding to a pinacol rearrangement of vicinal hydroxyl to form carbonyls in low pH environments [11]. The FNPs shown an excellent effect on H. pylori eradication because of their peroxidase-like activity, and FNPs with more C=O/C-O had greater affinity to bind the peroxidase substrates to enhance peroxidase-like activity and improve antibacterial effect. There are some related studies. Even some studies showed that functional groups such as carboxyl or carbonyl on the nanoparticles are necessary for imbuing them with peroxidase-like activity [15,16].

  1. It would be good to mention all stimuli in figure 6. For example, the temperate symbol is there but not designated by word.

Response: Thanks for the suggestion. The mean of temperate symbol in figure 6 is photothermal therapy (PTT), one of antibacterial photo-activated therapies. Some nanoparticles can produce localized heating (hyperthermia) when excited with visible-near infra-rad (NIR) light.

  1. Page 1, line 40:

"cationic AMPs display bactericidal action by disrupting the bacterial anionic cell membrane"

To support this statement, the following literature could be incorporated in this review.

The literature Title: Coordination-Assisted Self-Assembled Polypeptide Nanogels to Selectively Combat Bacterial Infection.

Response: We have cited this literature in the revised manuscript.

  1. Page 2, line 79:

"Non-Antibiotic Strategy Is Needed to Fight against Rapid"

The ‘I’ of the word ‘Is’ should be a small letter as 'is' and a similar issue with the word ‘against’. These should be taken care of.

Response: Thanks for careful checking. We have amended it.

  1. Page 2, line 83: "the abuse and misuse of antibiotics, bacteria have also developed a variety of drug-resistant mechanisms ". This statement is repeated many times in the manuscript (Abstract, Introduction, and in section 2 as well). It would be good to present it in a slightly different way.

Response: Thanks for the valuable suggestion. We have change it to “However, bacteria have also developed a variety of drug-resistant mechanisms to respond them”.

  1. Page 4, line 132: ‘In Vivo’ should be Italic.

Response: Thanks for careful check. We have amended it.

  1. In many places of the review, it appears that “Error! Reference source not found”. It should be considered to fix before publication.

Response: Thanks for reminding. We have fixed them.

Round 2

Reviewer 1 Report

The authors have addressed all of my comments/suggestions in their revised submission.